# Genome-Wide Analysis of *SPL*/miR156 Module and Its Expression Analysis in Vegetative and Reproductive Organs of Oil Palm (*Elaeis guineensis*)

**DOI:** 10.3390/ijms241713658

**Published:** 2023-09-04

**Authors:** Lixia Zhou, Rajesh Yarra

**Affiliations:** 1Coconut Research Institute, Chinese Academy of Tropical Agricultural Sciences, Wenchang 571339, China; 2Department of Plant and Agroecosytem Sciences, University of Wisconsin-Madison, Madison, WI 53706, USA; rajeshyarra@rediffmail.com

**Keywords:** *EgSPL*s, miR156 target sites, qRT-PCR, inflorescences

## Abstract

The *SPL* (SQUAMOSA-promoter binding protein-like) gene family is one of the largest plant transcription factors and is known to be involved in the regulation of plant growth, development, and stress responses. The genome-wide analysis of *SPL* gene members in a diverse range of crops has been elucidated. However, none of the genome-wide studies on the *SPL* gene family have been carried out for oil palm, an important oil-yielding plant. In this research, a total of 24 *EgSPL* genes were identified via a genome-wide approach. Phylogenetic analysis revealed that most of the *EgSPL*s are closely related to the Arabidopsis and rice *SPL* gene members. *EgSPL* genes were mapped onto the only nine chromosomes of the oil palm genome. Motif analysis revealed conservation of the SBP domain and the occurrence of 1–10 motifs in *EgSPL* gene members. Gene duplication analysis demonstrated the tandem duplication of *SPL* members in the oil palm genome. Heatmap analysis indicated the significant expression of *SPL* genes in shoot and flower organs of oil palm plants. Among the identified *EgSPL* genes, a total 14 *EgSPL*s were shown to be targets of miR156. Real-time PCR analysis of 14 *SPL* genes showed that most of the *EgSPL* genes were more highly expressed in female and male inflorescences of oil palm plants than in vegetative tissues. Altogether, the present study revealed the significant role of *EgSPL* genes in inflorescence development.

## 1. Introduction

The African oil palm (*Elaeis guineensis*) belongs to the palm family (Arecaceae) and is majorly cultivated as a source for palm oil. The palm oil is obtained from fruits of oil palm plants. The male and female inflorescences are formed separately in an alternating cycle on the same plant [1,2]. The inflorescences are continuously produced at the axil of each leaf along with the vegetative growth of oil palm plants [1,2]. The development of inflorescences occurs through different phases in a period of 2 to 3 years [3,4]. The successful development of inflorescence through differential phases is the key step for the formation of oil palm fruit for better oil yield [4]. However, differential gene expression also influence the development of oil palm inflorescences/flowers [4,5]. The differential expression of genes during the flowering are regulated at post-transcriptional level by various regulatory elements, including microRNAs [4]. To achieve the good yield of palm oil, the proper development of the oil palm fruit is needed, which is the source of palm oil. Prior to the fruit formation, growth and developmental stages of flowers are also most important. Recently, researchers identified the role of SPL genes in regulating the floral organ development by interacting with downstream genes that control the length and shape of inflorescence. So, it is noteworthy to identify the SPL genes in the oil palm genome and their specific expression in inflorescence for oil palm breeding.

Plant transcription factors (TFs) play a vital role in regulating the growth and development [6]. Among the plant transcription factors, the *SPL* (squamosa promoter binding protein, SBP-Box) gene family is an important plant specific transcription factor family known to regulate the growth and development of plants. The *SPL* gene was firstly identified in the cDNA library of *Antirrhinum majus* inflorescences as *SPL* proteins bind to the SQUAMOSA promoter of MADS-Box genes [7,8]. The *SPL* proteins contain a highly conserved SBP domain with 70 amino acid residues, including two tandem zinc fingers (Cys-Cys-His-Cys and Cys-Cys-Cys-His) and a C-terminus region with a nuclear localization signal (NLS) [9,10]. The NLS overlaps with the zinc finger structure to direct the SBP proteins into the nucleus for transcriptional regulation of downstream genes [11]. In recent years, various genome-wide studies have identified the occurrence of *SPL* gene members in various plants including 16 in sweet cherry [12], 23 in alfalfa [13], 18 in foxtail millet [14], 37 in trifolium [15], 15 in jatropha [16], 17 in *Arabidopsis* [17,18], 19 in rice [9], 56 in wheat [19], 24 in tartary buckwheat [20], 15 in pomegranate [21], 15 in tomato [22], 28 in poplar [23], 27 in apple [24], 18 in grape [25], and 17 in the orchardgrass genome [26]. However, no genome-wide studies have been carried out for identifying the *SPL* genes in the oil palm genome.

MicroRNAs (miRNAs) are endogenous non-coding RNAs known to suppress the expression of target genes at the post-transcriptional level [27,28]. Among all of the miRNAs, miR156 is highly conserved in plants and regulates the expression of *SPL* genes for transforming the plants from vegetative to reproductive phase [29,30,31]. The *SPL* genes regulated by miR156 are categorized into three groups including (i) *SPL*2, *SPL*9, *SPL*10, *SPL*11, *SPL*13, and *SPL*15 (promoting juvenile-to-adult vegetative transition and vegetative-to-reproductive transition); (ii) *SPL*3, *SPL*4, and *SPL*5 (promoting the floral meristem identify transition); and (iii) *SPL6* (function not yet known) [18]. Various studies also demonstrated the involvement of SPL genes for regulating physiological aspects related to growth and development, including leaf development, flower and fruit formation, and abiotic and biotic stress response. The leaf development is also regulated by *SPL* genes [32]: for example, *SPL*3 inhibits leaf primordia development; *SPL*9 and *SPL*10 control the leaf blade shape [18,33]. The grain size and shape in rice are regulated by *SPL*13 and *SPL*16 [34,35]. Moreover, *SPL* genes also play a vital role in abiotic and biotic stress response in various plants. Maize *SPL* genes are upregulated by cold, salt, and drought stress [36]. Downregulation of *SPL*8 improved drought and salt stress tolerance of transgenic alfalfa [37]. Enhanced salt tolerance of rice was also reported by knocking out the *SPL*10 gene in rice. Downregulation of three target genes *SPL*14, *SPL*11, *SPL*4 of *OsmiR*535 reduced the tolerance of rice to cold stress [38]. Spatiotemporal expression of alfalfa SPL genes under drought, salt stress, and biotic stress (methyl jasmonate) was also reported [13]. Previous studies also have shown that various *SPL* genes have miR156-binding sites [30,39]. Overexpression of miR156 in *Arabidopsis* downregulated the expression of *SPL* genes which have miR156 target sites [40]. Various researchers reported that floral organ development is regulated by *SPL*2 by activating the ASYMMETRIC LEAVES 2 gene of *Arabidopsis thaliana* [41]. The expression of *LEAFY* (*LFY*), *FRUITFULL* (*FUL*), and *APETALA1* (*AP1*) transcription factors in floral meristems is activated by the *SPL3*, *SPL4* and *SPL5* genes [42,43].

The aim of this study is to explore the oil palm inflorescence development mechanism via the SPL/miR156 module for genetic improvement and its utilization. Our genome- wide expression-profiling analysis of the SPL gene family in oil palm will provide a fundamental platform for candidate gene selection in oil palm biotechnology programs. In this study, we identified 24 *SPL* genes in the oil palm genome through a bioinformatics approach with the available oil palm genome sequencing data. This is the first ample report on gene structure, conserved motif analysis, chromosomal distribution, phylogenetic analysis, and duplication events of *EgSPL* genes in the oil palm genome. Heat map analysis from available transcriptome data of oil palm genome revealed the significant expression of *EgSPL* genes in shoot and floral tissues of oil palm plants. A total of 14 *EgSPL* genes possess the miR156 target sites. In addition, real time PCR analysis of *EgSPL* genes in vegetative and reproductive tissues revealed their significant expression in male and female inflorescences tissues. The expression levels of oil palm SPL genes in inflorescence provides some information to further study the biological functions in vegetative to floral transition and inflorescence development in this important oil-yielding crop. Altogether, our study provides the involvement of *SPL* genes during flower development in oil palm plants.

## 2. Results

### 2.1. SPL Genes Identification in E. guineensis Genome

A total of 24 *SPL* genes were identified in the oil palm genome through a genome-wide approach, and they were named as *EgSPL*1–*EgSPL*24. The gene IDs for all the oil palm *SPL* members are provided in Appendix A. The sequence information of CDS and protein for all the oil palm *SPL* members are provided in Appendix A. The length of the CDS for *SPL* genes ranges from 537 bp to 3282 bp, and the protein sequence length varies from 178 to 1093 amino acid residues (Appendix A).

### 2.2. EgSPL Gene Structural Features and Conserved Motif Analysis

To gain further insight into the structural features of oil palm *SPL* genes, we used Gene Structural Di*SPL*ay Server tool 2.0 (http://gsds.cbi.pku.edu.cn/, accessed on 1 June 2023) to analyze the exon/intron organization (Figure 1). Our analysis revealed the presence of a varied number of exons (2–12) among the identified 24 *SPL* genes. Highest numbers of exons (12) are possessed by *EgSPL*21, whereas *EgSPL*1, *EgSPL*9, *EgSPL*19, and *EgSPL*24 contained the lowest number of exons (Figure 1). Moreover, each *SPL* gene contained at least one intron, which varied among the *SPL* genes, indicating the functional role of introns in evolution. Further motif analysis revealed the existence of 10 motifs in *SPL* proteins which varied from 4 to 10 in each *SPL* protein (Figure 2). Moreover, motifs (1 and 2) related to SBP domains were found in all of the 24 oil palm *SPL* proteins (Figure 2). The presence of other motifs in addition to SBP domain motifs indicates the diverse functions of *SPL* genes.

### 2.3. miR156 Target Sites Prediction in Oil Palm SPL Genes

To identify the miR156 target sites in *EgSPL* genes, we searched the coding region and 3′-UTRs of all identified 24 *SPL* genes using the psRNATarget tool. We found that a total of 14 *EgSPL* genes (*EgSPL*2, *EgSPL*4, *EgSPL*5, *EgSPL*7, *EgSPL*8, *EgSPL*10, *EgSPL*11, *EgSPL*12, *EgSPL*13, *EgSPL*15, *EgSPL* 16, *EgSPL*17, *EgSPL*18, and *EgSPL*22) have miR156- binding sites either in coding or in 3′ UTR regions (Figure 3). The miR156-binding sites are present in the coding regions of 12 *EgSPL* (*EgSPL*2, *EgSPL*5, *EgSPL*7, *EgSPL*8, *EgSPL*10, *EgSPL*11, *EgSPL*13, *EgSPL*15, *EgSPL*16, *EgSPL*17, *EgSPL*18, and *EgSPL*22) members, whereas two *SPL*s (*EgSPL*4 and *EgSPL12*) contained miR156 sites in their 3′ UTR regions (Figure 3). Our results indicate that the regulation of *EgSPL* genes by miR156 is confined to a few genes among the identified 24 *EgSPL* genes.

### 2.4. Chromosomal Distribution and SPL Gene Duplication in Oil Palm Genome

The mapping of all identified 24 *EgSPL* gene across 16 chromosomes of the oil palm genome was also studied. A total of 20 *EgSPL* members were mapped on the chromosomes (1, 2, 3, 4, 7, 8, 10, 11, and 14), and the remaining 4 *SPL* genes were not mapped to any of the chromosomes. We did not find any *SPL* genes distributed on chromosome 5, 6, 9, 12, 13, 15, and 16. Chromosome 8 and chromosome 2 contained the highest number of *SPL* genes (Figure 4). Chromosomes 1, 3, 7, 10, and 14 had only one *SPL* gene. These results suggest the uneven distribution of *EgSPL* genes on chromosomes of the oil palm genome (Figure 4). Further, we used the Circos algorithm to learn the information on expansion of *SPL* gene duplications in the oil palm genome. We found that a total of 16 pairs of *SPL* genes were duplicated among the 16 chromosomes of oil palm. The duplicated pairs were located on the different chromosomes of the genome. Chromosomes 2 and 8 have the largest number of duplicated pairs compared to other chromosomes (Figure 5). Most of the chromosomes did not contain any duplicated pairs of *SPL*s, and duplication existed only on seven chromosomes. Our findings indicate the expansion or duplication of *SPL*s only exists through the duplicated blocks of the genome (Figure 5).

### 2.5. Evolutionary Relationship of Oil Palm, Rice, and Arabidopsis SPL Genes

We constructed the Maximum Likelihood tree for analyzing the evolutionary relation between oil palm *SPL* genes with the *SPL* genes of rice and Arabidopsis. Interestingly, oil palm *SPL* genes are distantly related with both rice and Arabidopsis *SPL* members (Figure 6). However, some of them are closely present in the clades where rice *SPL* members are grouped. Our results demonstrate that *EgSPL*s were separately evolved during evolution (Figure 6).

### 2.6. Expression Profiles of EgSPLs in Vegetative and Reproductive Tissues of Oil Palm

We examined the tissue-specific expression of 24 *EgSPL*s in vegetative (leaf, root, and shoot) and reproductive tissues (flower, fruit, and mesocarp (15, 17, 21, and 23 weeks old)) of oil palm plants using the available transcriptome data (SRR851096, SRR851071, SRR851067, SRR851108, SRR851103, SRR190698, SRR190699, SRR190701, and SRR190702). A total of two genes (*EgSPL*1, *EgSPL*3) in flower, a total of four genes (*EgSPL*2, *EgSPL*9, *EgSPL1*3, *EgSPL*16, *EgSPL*22) in shoot, and a total of two genes (*EgSPL*5, *EgSPL*6) in fruits are highly expressed compared to other *EgSPL* genes (Figure 7). However, a majority of the *EgSPL*s were expressed in flower and shoot tissues. All the identified 24 *SPL* genes were downregulated in root tissues (Figure 7). Based on the expression data, we hypothesize that the oil palm *SPL* gene family might play important roles in oil palm plant growth development, i.e., *EgSPL*1 and *EgSPL*3 in floral meristem development; *EgSPL*2, *EgSPL*9, *EgSPL1*3, *EgSPL*16, and *EgSPL*22 in shoot development; *EgSPL*5 and *EgSPL*6 in fruit development; *EgSPL*4 in leaf development.

### 2.7. Real-Time Expression Analysis of EgSPLs Containing miR156-Binding Sites

A quantitative real-time PCR experiment was performed to learn the relative expression of 14 oil palm *SPL* genes (containing miR156 sites) in various vegetative and reproductive tissues (male and female inflorescences) of oil palm plants. Our results demonstrate that expression of *EgSPL*02 and *EgSPL*12 is specifically confined to male inflorescences, whereas *EgSPL*07, *EgSPL*08, *EgSPL*18, and *EgSPL*22 are only confined to female inflorescences (Figure 8). A total of three genes (*EgSPL*05, *EgSPL*10, and *EgSPL*16) were only highly expressed in both the male and female inflorescences (Figure 8). A total of two genes, *EgSPL*13 and *EgSPL*17, were highly expressed in all the vegetative and reproductive tissues. Altogether, our results indicate the role of *SPL* genes in male and female inflorescence development in oil palm plants. We predict that *EgSPL*13 and *EgSPL*17 might play important roles in vegetative to reproductive phase transition, as both are expressed in vegetative and reproductive tissues of oil palm.

## 3. Discussion

*SPL*s are plant-specific transcription factors with a highly conserved SBP domain and are involved in regulating growth and development of plants. Various number of *SPL* genes were identified and characterized their expression in various crops including sweet cherry [12], alfalfa [13], quinoa [44], foxtail millet [14], wheat [45], pecan [46], soybean [47], Populas [48], and Jatropha [16]. However, some of the studies were carried out to characterize *SPL* gene expression during reproductive tissue development in various crops including pecan [46], Tartary buckwheat [20], rice [49], flowering cherry [50], Trifolium [15], and petunia [51]. However, to date none of the information is available on identification and characterization of *SPL* genes in oil palm during inflorescence development. In this study, we identified a total of 24 *SPL* genes in oil palm genome through a bioinformatics approach and dissected the expression of 14 *SPL*s in male and female inflorescences of oil palm plants through an experimental approach. The number of identified *SPL*s (24) in this study is fewer than the number of *SPL*s identified in sweet cherry (16 *SPL*s) [12], alfalfa (23 *SPL*s) [13], foxtail millet (18 *SPL*s) [14], and jatropha (15 *SPL*s) [16] and lesser than the number of *SPL*s identified in soybean (41 *SPL*s) [47], flowering cherry (32 *SPL*s) [50], wheat (56 *SPL*s) [45], Populus (33 *SPL*s) [48], and Trifolium (37 *SPL*s) [15] genomes. Present information on oil palm *SPL*s would elucidate the evolutionary process of *SPL*s across various plants.

The structural organization across the identified oil palm *SPL* genes showed that the variation in number of introns indicates the role of introns in *SPL* gene evolution. All the identified *SPL*s contained at least one intron and varied among them. Our results are consistent with the previous reports on identification of *SPL* genes in Jatropha [16], soybean [47], alfalfa [13], and sweet cherry [12]. The presence of the SBP domain is the key feature in the *SPL* gene family [9]. The motifs related to the SBP domain are found in all the identified *EgSPL* proteins. Conserved motif analysis revealed the occurrence of 10 motifs in oil palm *SPL* genes as reported in other plants including foxtail millet [14], alfalfa [13], and quinoa [44]. The post-transcriptional regulation of *SPL*s by *miR*156 determines fine-tuning functions of *SPL*s [52]. In our study, a total of 14 oil palm *SPL*s contained the miR156 sites, mostly in the coding regions and lesser in UTR regions. Our results are consistent with the previous reports on Jatropha, soybean, Medicago, and mustard *SPL* genes [16,47,53,54] indicating the conservation of *miR*156-mediated posttranscriptional regulation in plants. Phylogenetic analysis of 24 oil palm *SPL*s compared with *Arabidopsis* and rice *SPL*s suggested the diversification of oil palm *SPL* members among model-to-crop plants during the evolutionary process. The uneven distribution of 24 *SPL* genes on 16 chromosomes of oil palm also coincides with the previous reports on *SPL* gene family distribution in sweet cherry, alfalfa, and foxtail millet genomes [12,13,14].

*SPL* genes encode plant-specific transcription factors that play important roles in flower development, including vegetative to reproductive growth [55]. In our study, we analyzed the tissue specific expression of *EgSPL*s from the available transcriptome data and the majority of *EgSPL* genes expressed in flower and shoot. Our results are consistent with the previous reports on expression of *SPL* genes in blue berry [56]. Our qPCR data on expression of 14 *SPL* genes that contain miR156 sites revealed confined expression of some *SPL* genes in male and female inflorescences. The expression of miR156-targeted *EgSPL*02 and *EgSPL*12 was unique to male inflorescences, miR156-targeted *EgSPL*07, *EgSPL*08, *EgSPL*18, and *EgSPL*22 were unique to female inflorescences, and miR156-targeted *EgSPL*05, *EgSPL*10, and *EgSPL*16 were only highly expressed in both the male and female inflorescences, suggesting the involvement of *EgSPL* genes in inflorescence development. Our results are consistent with the previously reported involvement of miR156-targeted *SPL* genes in inflorescences [47,51].

## 4. Materials and Methods

### 4.1. Identification of SPL Gene Family in Oil Palm Genome

The known SPL protein sequences of Arabidopsis (16) and rice (19) were used as queries to retrieve the putative oil palm SPL family members against the oil palm (*E. guineensis*) genome database (http://palmxplore.mpob.gov.my/palmXplore/, accessed on 1 June 2023). Then, all the retrieved putative oil palm SPL family proteins were queried against the CDD (https://www.ncbi.nlm.nih.gov/Structure/bwrpsb/bwrpsb.cgi, accessed on 1 June 2023) and Pfam databases to confirm the occurrence of the conserved SBP domain. The length of each putative SPL gene coding sequence was also determined via Blastn search against the *E. guineensis* genome database.

### 4.2. Oil Palm SPL Gene Structure, Conserved Motif Analysis, and miR156 Target Site Prediction

The intron/exon structure analysis of 24 oil palm SPL gene structures was analyzed using Gene Structure DiSPLay Server (http://gsds.cbi.pku.edu.cn/, accessed on 1 June 2023). Further, the MEME tool was used for conserved motif analysis of all identified oil palm SPL protein members (http://meme-suite.org/tools/meme, accessed on 1 June 2023). The psRNATarget tool (http://plantgrn.noble.org/psRNATarget/?function=3, accessed on 1 June 2023) was used to predict the miR156 target site in oil palm full-length SPL gene sequences.

### 4.3. Phylogenetic Analysis, Duplication, and Chromosomal Distribution of Oil Palm SPL Genes

We generated the phylogenetic tree of oil palm SPL genes with the other known SPL genes of Arabidopsis and rice with the help of MEGA 7.0 tool by the Maximum Likelihood method, with a bootstrap value of 1000 replications [57]. We also investigated SPL gene members’ duplication events across the oil palm genome via the MCScanX tool with default parameters [58]. Further, we also mapped the distribution of 24 oil palm SPL genes on 16 chromosomes of oil palm from the oil palm genome database using TB tools software [59].

### 4.4. Plant Material and RNA Isolation

Healthy African oil palm (*Elaeis guineensis*) plants were grown in the field station of the Coconut Research Institute, Chinese Academy of Tropical Agricultural Sciences, Wenchang, China, and were all grown under institutional regulatory issues. The plant material used in this research work was collected by the corresponding author. To investigate the SPL gene expression analysis in different developmental stages, the samples including root, stem, leaf, male inflorescence, and female inflorescence were collected separately from the 6-year-old oil palm plants and quickly frozen in liquid nitrogen and stored at −80 °C for further experiments. Total RNA from root, stem, leaf, male inflorescence, and female inflorescence was isolated by following the method as described previously [60]. The yield, integrity, and purity of extracted RNA samples were quantified on Nanodrop and also electrophoresed on 1% agarose gel. The genomic DNA contamination was removed by treating the RNA samples with DNase I.

### 4.5. In Silico Expression Analysis of Oil Palm SPL Genes

To examine the SPL gene expression in various tissues of oil palm, the normalized RPKM values were retrieved from the available transcriptome data of six different tissues including leaf, root, flower, shoot, and mesocarp (15, 17, 21, and 23 weeks). The heatmap was generated to analyze the SPL gene expression in various tissues of oil palm using the pheatmap tool (https://cran.r-project.org/web/packages/pheatmap/index.html, accessed on 1 June 2023).

### 4.6. qRT-PCR Analysis of SPL Genes in Vegetative and Reproductive Tissues

To investigate the real-time expression of SPL genes in various vegetative and reproductive tissues, the real-time PCR expression analysis was performed using Mastercycler ep realplex4 Machine. The cDNA synthesis was carried out with the extracted RNA by using a MightyScript first-strand cDNA synthesis kit following the manufacturer’s instructions. The 2 × SYBR Green qPCR ProMix was used to carry out qRT-PCR reactions with Mastercycler. The oil palm SPL gene-specific primers (Appendix A) were designed by using the QuantPrime qPCR primer designing tool (https://quantprime.mpimp-golm.mpg.de/, accessed on 1 June 2023). Three biological and technical repeats were performed to determine the reliability of gene expression studies. The oil palm Actin1 gene was used as a housekeeping gene to check the relative expression of SPLs by employing the 2^−ΔΔCt^ method. The statistical significance was determined at *p* < 0.05 and *p* < 0.01 using ANOVA.

## 5. Conclusions

To the best of our knowledge, this is the first report on genome-wide analysis of *SPL* genes in oil palm. In the current investigation, a total of 24 *EgSPL*s were identified, and the expression of 14 *EgSPL* (containing miR156 sites) genes in vegetative and reproductive tissues of oil palm was analyzed. Moreover, detailed information on *SPL* gene structure, their miR156 target sites, motif composition, chromosomal location, and phylogenetic analysis was also reported. Furthermore, the unique expression of *EgSPL*s (containing miR156 sites) in oil palm inflorescences was also revealed via qPCR analysis, predicting their putative role in male and female inflorescence development of oil palm.

## Figures and Tables

**Figure 1 ijms-24-13658-f001:**
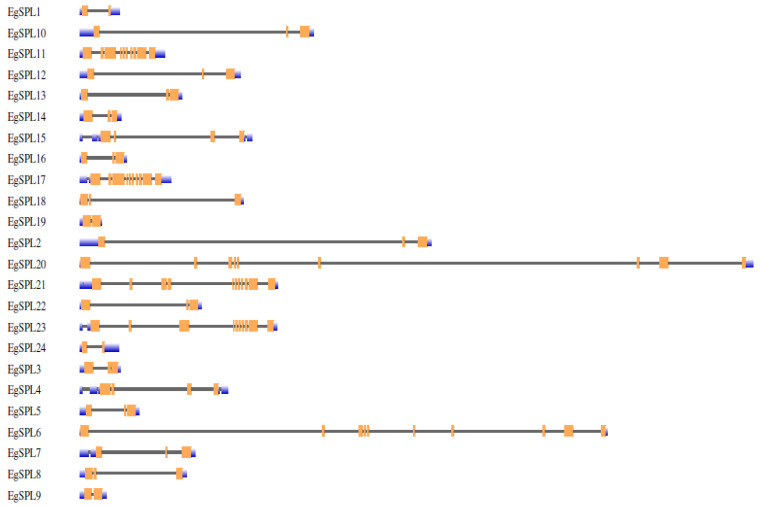
Intron-exon organization of 24 *EgSPL* genes. Coding sequences (CDS), untranslated regions (3′ and 5′ UTRs). Intron regions are represented by orange, blue, and black color blocks, respectively, in the schematic presentation.

**Figure 2 ijms-24-13658-f002:**
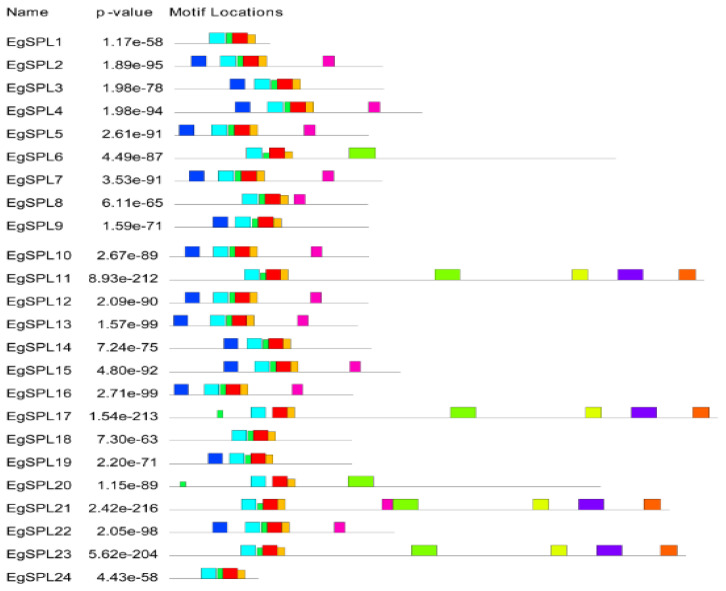
Motif analysis of *EgSPL* proteins using the MEME tool. Presence of 1–10 motifs in *EgSPL* proteins. Each motif is represented with different color. The abundance of each amino acid in their motifs is represented by sequence logo of each motif.

**Figure 3 ijms-24-13658-f003:**
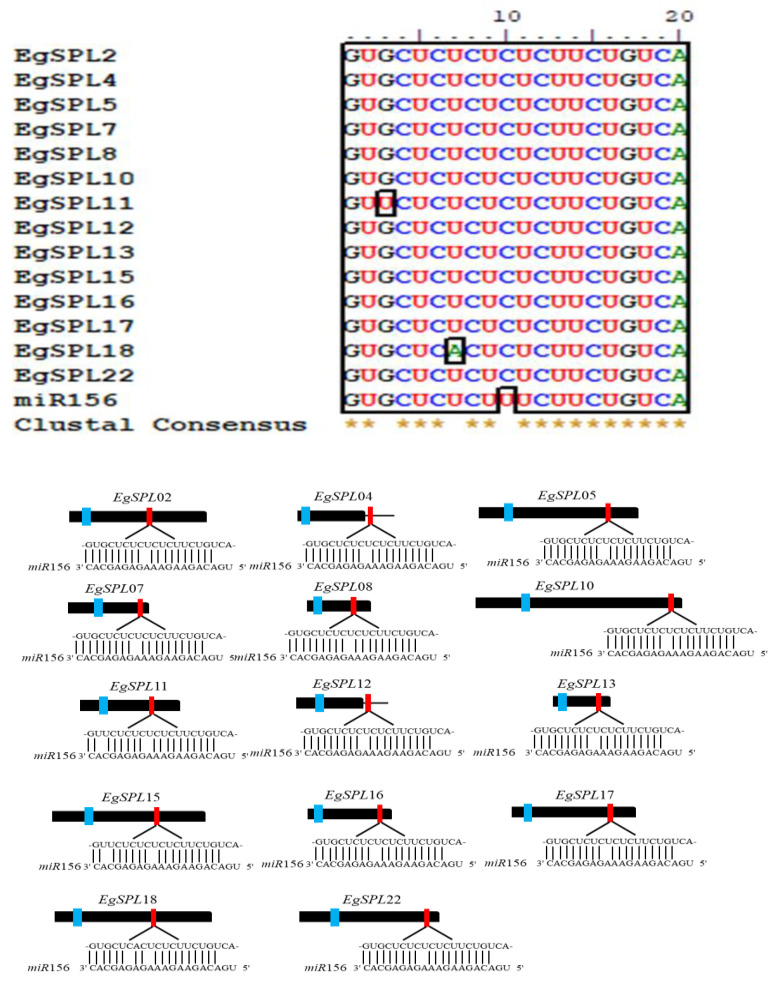
Prediction of miR156 target sequences of 14 *EgSPL*s through psRNATarget tool. Multiple alignment of miR156 complementary sequences with their targets (upper panel). Lower panel diagram represents the gene structure of 14 *EgSPL*s. The black boxes represent CDS regions, the black lines represent 3′ and 5′ UTR regions, the blue boxes represent the SBP domains, and the red boxes represent miR156 target sites of *EgSPL* transcripts. In the expanded regions, the sequence direction of *EgSPL* is from 5′ to 3′ and the miR156 sequence direction is from 3′ to 5′.

**Figure 4 ijms-24-13658-f004:**
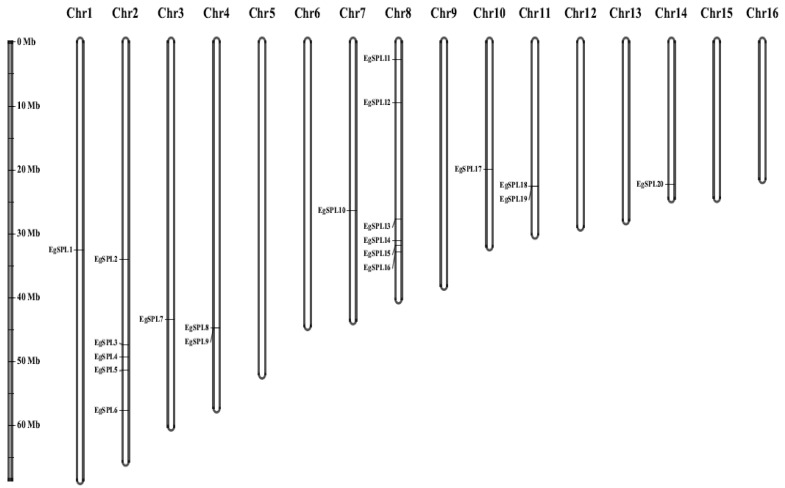
Uneven distribution of *EgSPL* genes on 16 chromosomes of oil palm genome. Chromosome numbers from 1–16 are marked on the top portion of each chromosome. The length of the oil palm chromosomes is represented with the vertical greyscale on the left side.

**Figure 5 ijms-24-13658-f005:**
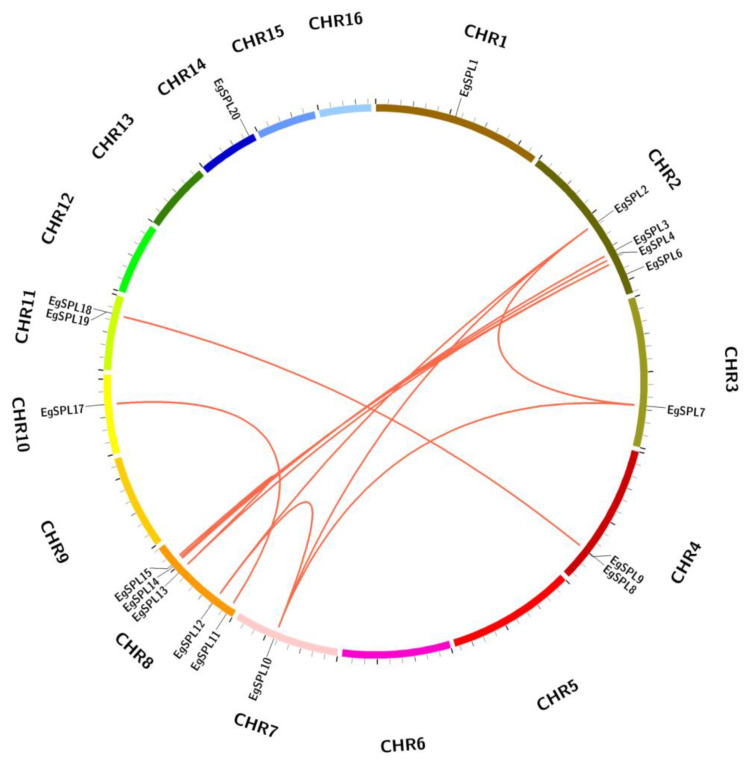
*EgSPL* gene duplication events in oil palm genome. The duplicated gene pairs are linked by the red lines inside the circle view as revealed by MC ScanX tool. Each chromosomal block is represented by a different color.

**Figure 6 ijms-24-13658-f006:**
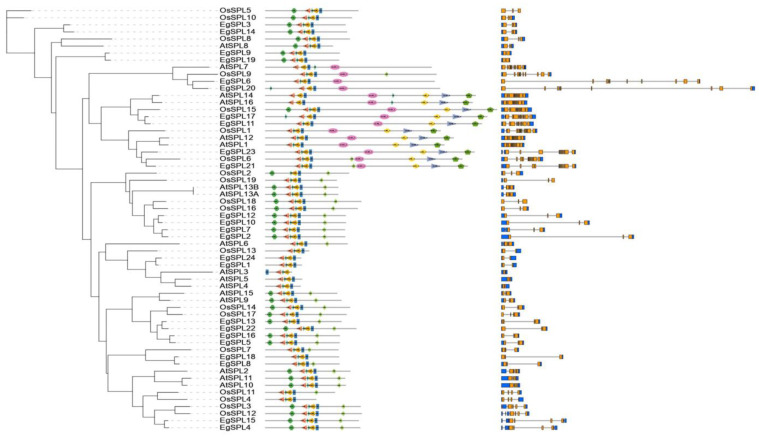
Phylogenetic analysis of *EgSPL* gene family with rice and Arabidopsis *SPL* family genes. Domain composition (**middle**) and gene structural organization (**right**) of oil palm, rice, and Arabidopsis *SPL* members are also represented in the illustration.

**Figure 7 ijms-24-13658-f007:**
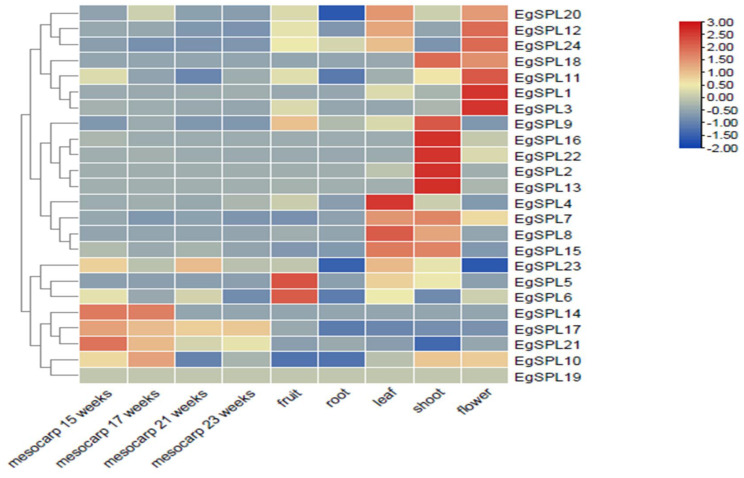
Tissue-specific expression profile of *EgSPL* genes in vegetative (leaf, root, and shoot) and reproductive tissues (flower, mesocarp, and fruit) based on the available transcriptome data of oil palm. SRR190698 represents transcriptome of mesocarp (15 weeks); SRR190699 represents transcriptome of mesocarp (17 weeks); SRR190701 represents transcriptome of mesocarp (21 weeks); SRR190702 represents transcriptome of mesocarp (23 weeks); SRR851108 represents transcriptome of flower; SRR851067 represents transcriptome of fruit; SRR851096 represents transcriptome of leaf; SRR851071 represents transcriptome of root; SRR851103 represents transcriptome of shoot.

**Figure 8 ijms-24-13658-f008:**
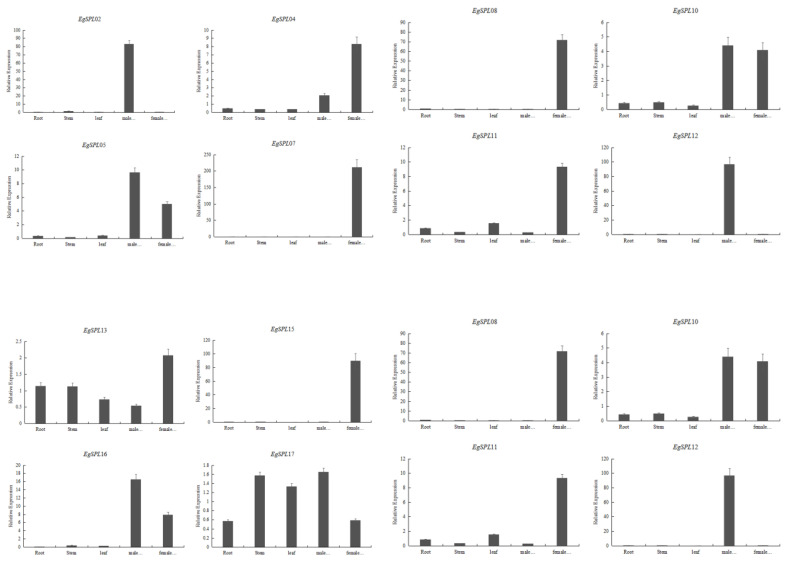
Relative expression of 14 *EgSPL* transcription factors (containing miR156 sites) in vegetative and reproductive tissues. Data represent the mean ± SE of three replicates.

## Data Availability

All data is available upon requesting the corresponding author.

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
