# Peer review of "Genome-Wide Analysis of *SPL*/miR156 Module and Its Expression Analysis in Vegetative and Reproductive Organs of Oil Palm (*Elaeis guineensis*)"

_ijms, 2023, doi:10.3390/ijms241713658_

Round 1
Reviewer 1 Report
The study revealed the gene expression responsible for the inflorescences and flowers development in oil palm, an important crop. This study are justified as allow the gene modification in the plant breeding process.
I found the study properly designed and the results are clearly presented.
My main concern is related to the study aim.
In fact , the objectives of the study are not clearly stated.The authors just inform what they did. Try to rephrase the aim of study paragraph to underline, why the study were conducted?
May be the hypothesis can be formulated?
Please also try to explain the importance of the research.
Author Response
Response to Reviewer Comments
The study revealed the gene expression responsible for the inflorescences and flowers development in oil palm, an important crop. This study are justified as allow the gene modification in the plant breeding process. I found the study properly designed and the results are clearly presented.
Response: Thanks for the reviewer comments
My main concern is related to the study aim. In fact , the objectives of the study are not clearly stated. The authors just inform what they did. Try to rephrase the aim of study paragraph to underline, why the study were conducted?May be the hypothesis can be formulated? Please also try to explain the importance of the research.
Response: Thanks for the reviewer comments and suggestions. Now, we provided the information in last paragraph of introduction section, as suggested by reviewer.
The aim of the study is to explore the oil palm inflorescence development mechanism via SPL/miR156 module for genetic improvement and its utilization. Our genome wide and expression profiling analysis of SPL gene family in oil palm will provide a fundamental platform for candidate gene selection in oil palm biotechnology program. In this study, we identified 24 SPL genes in oil palm genome through bioinformatics approach with the available oil palm genome sequencing data. This is the first ample report on gene structure, conserved motif analysis, chromosomal distribution, phylogenetic analysis and duplication events of EgSPL genes in oil palm genome. Heat map analysis from available transcriptome data of oil palm genome revealed the significant expression of EgSPL genes in shoot and floral tissues of oil palm plants. A total of 14 EgSPL genes possess the miR156 target sites. In addition, real time PCR analysis of EgSPL genes in vegetative and reproductive tissues revealed their significant expression in male and female inflorescences tissues. The expression levels of oil palm SPL genes in inflorescence provides some information to further study the biological functions in vegetative to floral transition and inflorescence development in this important oil yielding crop. Altogether, our study provides the involvement of SPL genes during the flower development in oil palm plants.
Reviewer 2 Report
The authors identified the SPL homologs in oil palm and looked at expression of these genes. What is noteworthy is the relatively specific expression of many of these genes. This work is descriptive, but doesn't address function or action of these loci. What are issues specific to oil palm that make examination of these genes worthwhile? Why was this work undertaken? The presentation and analysis are solid, but I am left with the question, "Why did you do this?"
Fine.
Author Response
Response to reviewer comments
The authors identified the SPL homologs in oil palm and looked at expression of these genes. What is noteworthy is the relatively specific expression of many of these genes. This work is descriptive, but doesn't address function or action of these loci. What are issues specific to oil palm that make examination of these genes worthwhile? Why was this work undertaken? The presentation and analysis are solid, but I am left with the question, "Why did you do this?"
Response: Thanks for the reviewer comments and suggestions. Now we provided the information in introduction , as suggested by reviewer.
To achieve the good yield of palm oil, it’s needed for the proper development of the oil palm fruit, which is the source of palm oil. Prior to the fruit formation, growth and developmental stages of flowers are also most important. Recently, researchers identified the role of SPL genes in regulating the floral organ development by interacting with downstream genes that control the length and shape of inflorescence. So, it’s noteworthy to identify the SPL genes in oil palm genome and their specific expression in inflorescence for oil palm breeding.
The aim of the study is to explore the oil palm inflorescence development mechanism via SPL/miR156 module for genetic improvement and its utilization. Our genome wide and expression profiling analysis of SPL gene family in oil palm will provide a fundamental platform for candidate gene selection in oil palm biotechnology program. In this study, we identified 24 SPL genes in oil palm genome through bioinformatics approach with the available oil palm genome sequencing data. This is the first ample report on gene structure, conserved motif analysis, chromosomal distribution, phylogenetic analysis and duplication events of EgSPL genes in oil palm genome. Heat map analysis from available transcriptome data of oil palm genome revealed the significant expression of EgSPL genes in shoot and floral tissues of oil palm plants. A total of 14 EgSPL genes possess the miR156 target sites. In addition, real time PCR analysis of EgSPL genes in vegetative and reproductive tissues revealed their significant expression in male and female inflorescences tissues. The expression levels of oil palm SPL genes in inflorescence provides some information to further study the biological functions in vegetative to floral transition and inflorescence development in this important oil yielding crop. Altogether, our study provides the involvement of SPL genes during the flower development in oil palm plants.
Round 2
Reviewer 2 Report
In my previous review, I requested that the authors explain the ramifications of this work in oil palm. SPL genes carry out a diverse array of functions. The authors have focussed on the ability of some SPLs to promote phase transition and plant reproduction. I want the authors to discuss ALL of the SPL activities and explain how each of the SPLs might have other activities. Review the expression data to narrow possible activities individual SPLs may have. The resubmission has not done this. Analyze your data.
Fine.
Author Response
Response to Reviewer Comments
1) In my previous review, I requested that the authors explain the ramifications of this work in oil palm. SPL genes carry out a diverse array of functions. The authors have focussed on the ability of some SPLs to promote phase transition and plant reproduction. I want the authors to discuss ALL of the SPL activities and explain how each of the SPLs might have other activities.
Response: Thanks for the reviewer comments. We tried our best to provide all the SPL gene activities related to other than plant reproduction in introduction section as stated below.
“Various studies also demonstrated the involvement of SPL genes for regulating physiological aspects related to growth and development, including leaf development, flower and fruit formation, abiotic and biotic stress response. The leaf development is also regulated by SPL genes [32] such as SPL3 inhibits leaf primordia development; SPL9 , SPL10 controls the leaf blade shape [33,34]. The grain size and shape in rice is regulated by the SPL13 and SPL16 [35,36]. Moreover, SPL genes also play vital role in abiotic and biotic stress response in various plants. Maize SPL genes are upregulated by cold, salt and drought stress [37]. Down regulation of SPL8 improved drought and salt stress tolerance transgenic alfalfa[38]. Enhanced salt tolerance of rice was also reported by knocking out the SPL10 gene in rice. Down regulation of three target genes SPL14, SPL11, SPL4 of OsmiR535 reduced the tolerance of rice to cold stress[39]. Spatio temporal expression of alfalfa SPL genes under drought, salt stress and biotic stress (methyl jasmonate) was also reported [40]”
2)Review the expression data to narrow possible activities individual SPLs may have. The resubmission has not done this. Analyze your data.
Response: Thanks for the reviewer comments. We tried our best to provide the information on individual SPLs. However, its not possible to predict the each oil palm SPL gene role without functional studies. We gave information as stated below in results section.
In the section “Expression profiles of EgSPLs in vegetative and reproductive tissues of oil palm”
“Based on the expression data, we are assuming that oil palm SPL gene family might play important role in oil palm plant growth development i.e. EgSPL1 and EgSPL3 in floral meristem development; EgSPL2, EgSPL9, EgSPL13, EgSPL16, EgSPL22 in shoot development; and EgSPL5, EgSPL6 in fruit development, EgSPL4 in leaf development.”
In the section “Realtime expression analysis of EgSPLs containing miR156 binding sites”
We are predicting that EgSPL13 & EgSPL17 might play important roles in vegetative to reproductive phase transition, as both are expressed in vegetative and reproductive tissues of oil palm.
Round 3
Reviewer 2 Report
Thank you for editing the ms. to provide more context for this work.
Copy-editing should clear up any ambiguities.